# Organizational and Personal Factors That Boost Innovation: The Case of Nurses during COVID-19 Pandemic Based on Job Demands-Resources Model

**Ariana Moreno Cunha** [1,*], **Carla Susana Marques** [2] and **Gina Santos** [3]

1 Hepatology Day Care Hospital, Centro Hospitalar Trás-os-Montes e Alto Douro (CHTMAD), 5000-508 Vila Real, Portugal

2 Department of Economics Sociology and Management, University of Trás-os-Montes e Alto Douro (UTAD) & CETRAD Research Centre, 5000-801 Vila Real, Portugal; smarques@utad.pt

3 Lusófona University 4000-098 Porto & CETRAD Research Centre, 5000-801 Vila Real, Portugal; gina.santos@utad.pt

* Correspondence: ariana-moreno@hotmail.com

**Abstract:** Since 2019, the world has been experiencing a pandemic period due to the COVID-19 virus, which has brought the need for organizations in general, healthcare organizations and their professionals in particular, to focus on innovation as a way to fight an utterly unknown virus. Thus, this study aims to understand how nurses and their personal factors (stress, anxiety, work engagement, organizational support) impact their innovative behaviour and innovation outputs, contributing to innovation in the current pandemic period through changes in thoughts, values, behaviours and relationships among healthcare professionals and their organizations. For this purpose, the Job Demands-Resources model was used as a reference, and the measurement instrument was applied to 738 nurses working in healthcare units in Portugal. Therefore, it was found that the nurses' personal factors have a positive effect on the nurses' innovative behaviour and innovation outputs, with the innovative behaviour having the most significant impact on innovation outputs, which will benefit healthcare organizations and the healthcare provided to patients during the pandemic, through innovative behaviours and products. It is also possible to understand how the available resources and the demands imposed on nurses interfere with their innovative behaviour (Job Demands-Resources model).

**Keywords:** stress; anxiety; work engagement; organizational support; innovative behaviour; innovation outputs; job demands-resources model; nurses; COVID-19

## 1. Introduction

Innovation is relevant not only in management but also in healthcare, more specifically in the nursing area. The approach to innovation is related to other current themes such as the personal factors involving nurses (stress, anxiety and work engagement) and the organizational support, which can help us understand these professionals' response to the innovative behaviour and inherently to the innovation outputs. On the one hand, in the current context and global terms, there is a need for the international community to come together to find ways to combat COVID-19. On the other hand, it is also essential to understand how this pandemic will condition healthcare teams. Human capacities, healthcare professionals and available resources are being tested, forcing professionals to innovate and at the same time manage stress and anxiety, which may or may not influence the results obtained.

In addition to substantiating the dimensions under analysis with the Job Demands-Resources model to sustain the results obtained, we also sought to review the respective model, assuming innovative behaviour as a positive factor that will contribute to the innovation results within the organization.

The meaning of stress varies from person to person, taking into account the different situations in which they find themselves, and can be generically described as being a non-specific response of the body to a given crisis, and is therefore not something that can be prevented [1]. However, according to the study of [2], as long as nurses who work with COVID-19 patients are appropriately monitored at the psychological level at an early stage, they do not report short-term psychological changes, although regular assessment is necessary. Thus, with this study, we tried to understand whether the stress experienced by nurses in Portugal influences their behaviours or not.

Similarly, as anxiety is related to mental and physiological phenomena, it is necessary to distinguish whether it is a standard or pathological behavioural condition, preventing or reducing the normal development of adaptive behaviours [3], such as work-related behaviours. Therefore, in the study by [4], it is mentioned that anxiety related to the COVID-19 pandemic is frequent in nurses, which may affect their well-being and work performance. It is necessary to find elements that can help maintain mental health and, at the same time, reduce nurses' work-related anxiety during this pandemic. Hence, our research is crucial to understand the role of nurses' anxiety in their innovative behaviour.

According to the theoretical research of [5], work engagement can be defined as a psychological, affective, emotional, motivational and social condition that involves a positive mental and motivational state. For the same authors [5], work engagement is not only about persevering over time but is also being described by the vigour, dedication, absorption that surround it and by a sense of achievement that is always work-related. Therefore, it is of enormous relevance in a pandemic moment such as the one we are currently experiencing. The literature review of [6] also mentions that nurses' work engagement demonstrates the commitment these professionals have with their profession while facing different challenges related to the pressure on healthcare systems during the COVID-19 pandemic. Not only is the pandemic a pressure in itself, but also the increased demand for quality healthcare, the patients' expected outcome, not excluding the shortage of nursing professionals in different healthcare organizations [6]. Thus, our work also aims to understand the role of work engagement in nurses' innovative behaviour and innovation outputs.

Regarding personal factors in general and stress, anxiety or work engagement in particular, it should be noted that there are studies (e.g., [7–11]) that address these themes and relate them, but the number of studies investigating these dimensions applied to nurses is rare (e.g., [2,4,6,12–14]), especially in a pandemic period.

Organizational support is related to how employees perceive their work is valued by the organizations and their concern for their well-being [15]. Thus, during the COVID-19 era, this aspect becomes even more critical to the extent that nurses need to understand the support of the organization where they work by recognizing their psychological needs, by providing the support needed to their professionals [16]. Therefore, it is relevant to understand the role of organizational support in nurses' innovative behaviour and innovation outputs so that there is an evolution in innovation and the relationship between professionals and the health institutions where they work.

Innovative behaviour should be treated carefully because innovative results can be expected from employees, leading them to behave towards the production of new products, services or work processes, and this behaviour can be affected by several factors [17,18]. Therefore, employees are more likely to adopt innovative behaviour if they are given freedom and autonomy [19], allowing them to demonstrate behaviours through which they generate or adopt new ideas, subsequently making extra efforts to execute them, as mentioned more recently by [20]. According to the study of [21], during COVID-19, a change in healthcare organizations is assumed to be derived from changes in the nurses' behaviour that has become more innovative in emergencies, such as a pandemic. However, the same authors mention that some organizations do not undergo significant changes due to the low adherence of their employees. Thus, we intend to determine whether the innovative behaviours of nurses in Portugal are significant during COVID-19.

The Job Demands-Resources model interconnects the dimensions investigated in this study either directly or indirectly. The study of [22] initially included only four primary constituents in this model: work demands, work resources, burnout, and engagement, but the model allowed other characteristics to be associated as relevant in explaining burnout, depending on the specific professional group under study. Later, [23] made a more comprehensive proposal of the Job Demands-Resources model, assuming leadership as an integral part of the model, and on the other hand, [24] considered the relevance of the leader's feedback on innovative behaviour and work engagement. In this sense, and to the extent that engagement is positively related to innovative behaviour [24,25], we believe it would be entirely relevant to study innovative behaviour and innovation outputs based on the Job Demands-Resources model. For us, innovative behaviour influences the professional's attitude within the organization and innovation outputs that produce improvements and success for organizations.

Moreover, according to the literature review, we did not find studies linking all the dimensions that we intend to address in this research.

It should also be noted that there are already studies that relate, in an isolated and more or less exhaustive way, the relationships between the dimensions under investigation (e.g., [24,26–29], there being, however, a lack of studies in the healthcare sector, in general, and in a COVID-19 pandemic situation, in particular. Therefore, we also try to contribute to science by understanding the above relationships, filling the gap of correlating all dimensions and demonstrating how personal and organizational factors influence innovative behaviour. In addition, it is also necessary to realize that the studies conducted so far have not included, in their entirety, nurses working in hospitals during a pandemic, which in itself makes this research something innovative and original. COVID-19, in itself, invited healthcare professionals to find new ways of working and combating their difficulties. Our study also intends to demonstrate how these professionals behave in innovation, even when facing so many doubts and fears.

It should be noted that there is an increasing need to conduct studies in the healthcare area, specifically on nurses. This class is subject to greater demands at work since they work in shifts, are constantly in mourning, and are not well paid; thus, there is their professional, personal, family and social lives are substantially compromised. In this sense, the JDR theory helps us to understand the implications of these demands, which impact or influence, or not, innovative behaviour and innovation outputs. Thus, we highlight, at the level of individual demands, the dimensions of stress and anxiety, which are based on the SDT theory; of work engagement, which is based on the B&B theory; and, at the organizational level, we rely on the SET theory based on the dimension of organizational support.

The originality of this study is related to the fact that we intend to provide a broad understanding of the relationship between nurses' personal and organizational factors during the COVID-19 pandemic, and the innovative behaviour of these professionals and the innovation outcomes, applying the Job Demands-Resources model. We believe that this study is one of the first to assess the results of innovation (innovative behaviour and innovation outputs) of nurses during the pandemic, since it forced these professionals to develop and implement a strategy in which they could, as a team, promote innovative behaviour and generate innovation outputs, and, at the same time, assess the relationship between nurses' personal factors and nurses' innovations.

In this study, the relationship between the personal factors influencing nurses and their innovative behaviour during the COVID-19 pandemic was analyzed through a literature review related to the dimensions under analysis and the distribution of a questionnaire among nurses, whose data were subsequently analyzed. Therefore, we found that personal factors such as stress, anxiety, work engagement, and organizational support positively influenced nurses' innovative behaviour and innovation outputs. Therefore, even during a pandemic, which destroys so much around them, nurses can find themselves, together with their organizations, their personal and professional characteristics, time and space to innovate and bring something positive to the world.

This article is divided into six sections. In the following section, we set out the theoretical foundations of the topics addressed that support the research hypotheses that will give rise to the proposal of the conceptual research model of this study. In the next section, we develop the methodology used in this research. In the fourth section, we present the main results, namely the characterization of the sample, the validation of the measurement instrument and the structural equation model based on the proposed conceptual research model. In the fifth section, a brief discussion of the results is made, and in the last section, the main conclusions and future recommendations are presented.

## 2. Literature Review

### 2.1. Innovative Behaviour

Innovation is an organizational process that comprises three main elements: idea generation, promotion and realization [30]. Therefore, this is a process that requires the full involvement of employees to maintain continuity, sustainability, the achievement of competitive advantages, as well as the growth and success of the organization, being necessary to increase the motivation of individual employees and the team in general [31,32], giving relevance to aspects related to performance, learning and innovation, so the organization is thriving and competitive [33].

Individual innovative behaviour is directed towards the projection and use of new and profitable ideas, methods, products or behaviours [34]. On this type of behaviour, managers motivate their employees to demonstrate innovative behaviour [35], so this type of conduct is affected by several factors such as the leader and their support for innovation, perspectives of management roles, length of a professional career and the pursuit of problem-solving [17].

High-quality relationships in an organization positively impact innovative behaviour [36], with employees being more likely to adopt innovative behaviour because they are given the freedom and autonomy to bridge the gaps they found [19]. This is only possible if there is a quality relationship with the leader, since employees will accept challenging tasks, which, together with the leader's support and the incentive to deal with risks, allows for the creation of a beneficial environment for the emergence of innovative behaviour [37].

The Job Demands-Resources model argues that working conditions are of two types: job demands and job resources, where resources refer to the physical, psychological, social, or organizational aspects of work that can enable the achievement of the work goals, reduce job demands, and stimulate personal growth and development [22]. This model also integrates several aspects that contribute to employee work engagement, interpreting the demands, labour and individual resources necessary for this engagement to occur, and considering the well-being, performance and innovative behaviour as a consequence of these interactions [8,38,39].

It was addressed on the study of [39] that the relationship between work engagement and innovative behaviour, based on the Job Demands-Resources model and other theories. Therefore, concerning work resources at the individual level, which refers to job characteristics and personal resources, they address the B&B (Broaden and Build) theory, which states that more positive employees, in terms of resilience, self-esteem and effectiveness, have more confidence to make more challenging choices, following non-traditional approaches, in which employees must take risks to obtain innovative behaviour's [39].

Therefore, we consider that innovative behaviour is an essential dimension of the Job Demands-Resources model since several authors have already directly related it to work engagement and leader feedback [24] or to workload and work engagement [40] or even to individual resources [39]. These associations also exist in other studies (e.g., [8,23]) that report the relationship between high resources, motivation, employee commitment and excellent performance, translated by work engagement, which will lead to employees' innovative behaviour, in line with our conceptual model based on the Job Demands-Resources model.

According to [41], the innovative behaviour of healthcare professionals plays a crucial role in innovation and the determinants of innovative behaviour at work are not yet fully

understood, so there is a need to understand the organizational factors that influence this same innovation. The same authors state that the results of their study support the idea of the importance of the organizational climate and openness to innovation in the emergence of the healthcare professionals' innovative behaviour, which will depend on the organization's acceptance of innovation itself.

Reference [42] mentions that nurses' innovative behaviour is related to their participation in improving their work and adopting a plan to improve their care by developing new nursing behaviours. Therefore, innovation is relevant in eliminating outdated protocols or improving existing care plans to create new work behaviours [43].

As such, many managers encourage nurses to innovate through the use of products, services, technologies, and methods, which improve outcomes and productivity at work by integrating innovation into the daily routines of these professionals [44].

However, according to several Chinese studies, nurses are shown to have medium or low levels of innovative behaviour when compared to other professionals [45], which may be explained by their need for help and support from their managers in the process of discovering the resources needed to implement new ideas and behaviours [17].

*2.2. Innovation Outputs*

Deep and detailed research was conducted by [20] in which they developed a model of employees' innovative behaviour that integrates several factors included in several scales. Through the more detailed evaluation of these scales, these authors performed a compilation of only a few items, using only those that were tested and proved to be most relevant, and no longer have only one scale that speaks of innovative acts of employees but also started to consider innovation outputs.

Insofar as there is much literature about innovation and employees' innovative behaviour, we also find different definitions that may mix concepts, where sometimes innovation outputs are confused with other ideas, when in reality these are only achieved when new ideas are implemented in an organization, through changes achieved in a particular product, service or process [20].

Therefore, when building up their theoretical model of the employee's innovative behaviour, [20] inserted the six dimensions concerning the innovative behaviour (idea generation; idea search; idea communication; implementation of activities; involvement of others in these activities; overcoming obstacles). They also insert the innovation outputs construct (considered to be the results achieved) and the contextual influences that the authors considered to be more relevant in the employee's innovative behaviour, such as work environment, cultural support, organizational support and management support, thereby assuming that management support, in general, is a relevant factor in the innovative attitudes of employees, in the case of a nurse's innovative behaviour [20].

Although [46] have also created a scale with several distinct dimensions regarding innovative behaviours and various other authors have addressed several measures of employees' innovative behaviour, as well as the measurement of innovation outputs, it was [20] who allowed us to address this issue in more depth in this study.

In this way, the following research hypothesis is formulated:

**Hypothesis 1 (H1).** *Nurses' innovative behaviour promotes innovation outputs.*

*2.3. Stress and Anxiety*

Stress is a reaction that individuals have when they are exposed to pressures, demands and efforts above the expected level, which may become an undesirable situation that affects people at the physical and mental levels [47]. When it occurs in the workplace, it makes employees demand more of themselves than before to cope with the work demands [48]. If stress is excessive, it generates anxiety in individuals, which leads to a reduction in employee performance [47].

Although anxiety has become present in most phases and situations of modern life throughout the world, it becomes more evident when there are situations of high uncertainty [49] or when individuals are simply unsure whether they will have the ability to cope with a problem that is potentially threatening to their social position [50].

The continuous physical or emotional stress and the workload to which nurses are subjected can lead these professionals to show signs of distress and depression, which may negatively affect their psychological well-being [40,51,52]. Therefore, hospital managers should be more alert to the leading causes of work-related stress through occupational health so they can protect their nurses' physical and psychological health [12].

As several authors negatively correlate anxiety with motivation, nurses' professional performance may be affected by the anxiety they experience in their daily lives [53–55].

On the other hand, the results of [36] presented a positive relationship between work-related anxiety and employees' innovative behaviour, mainly when equity levels in the organization were low, strengthening the idea that anxiety and innovative behaviour may generate innovative ideas.

The approach of [9] to the Job Demands-Resources model state that the greater the tension at work to which nurses are exposed, the less able they are to manage stress, anxiety and organizational demands, thus work stress may be related to bureaucracy, role conflicts, task repetition or excessive work pressure [56], causing the nurses' opportunity to grow and develop to become negatively involved, thus causing extreme stress [9]. Therefore, it is understandable that [8] associate the Job Demands-Resources model with work demands, ranging from the unfavourable physical environment to emotionally demanding interactions, such as nurses' relationship with patients, families and other professionals around them, in healthcare institutions.

The demands mentioned in the Job Demands-Resources model, such as stress and anxiety factors, are related to professional resources at the individual level, according to [39], and are also related to Self-Determination Theory (SDT), which states that when an employee makes a choice and has autonomy and control over their work, self-determination is achieved [57]. According to [8], demands, such as stress factors, may not be harmful, as long as the resources are also high, contributing to employee engagement.

However, suppose nurses can cope with their work-related stress and remain highly engaged in their work. In that case, they are more likely to feel fulfilled, as emotional exhaustion related to work stress directly affects nurses' professional achievement and work engagement [14].

Thus, the need to address this topic becomes unquestionable when health professionals face daily challenging situations, increasing the interest in understanding how stress and anxiety in nurses' impact their work and innovation during the COVID-19 stage.

Based on the above knowledge, the following research hypotheses are presented:

**Hypothesis 2 (H2).** *Stress influences nurses' innovative behaviour.*

**Hypothesis 3 (H3).** *Anxiety influences nurses' innovative behaviour.*

*2.4. Work Engagement*

Work engagement can be described as a situation in which employees regard their work as meaningful, positive and rewarding, characterized by vigour, dedication and absorption, to the extent that there are facilitating factors for the development of work engagement such as social support; job performance; personal resources; organizational resources and issues; and resilience [58].

Work engagement allows employees to use their cognitive resources to find new perspectives, information and knowledge, which combined let them conceive new ideas [59]. However, for [60], work engagement is about resources and the effort that the employee is willing to invest at work.

Therefore, work engagement is positively affected by employee emotional motivation [61] and, as positive organizational behaviour, it has a significant effect on employee behaviour [62]. However, for work engagement to manifest itself, there must be professional and personal resources, which independently or together increase employee engagement in their work within the organization [8].

The study of [8] states that the Job Demands-Resources model addresses the risk factors associated with work stress (demands and job resources) and motivation and its potential contribution to work engagement. Therefore, for this to occur in organizations, personal resources are needed to motivate and engage employees in activities that satisfy their unique needs or moderate stress, acting as a source of energy for work engagement and overcoming daily work challenges [11].

Work engagement is significantly related to employees' performance and their innovative behaviour, as in this way they show more interest and enthusiasm towards the work, being more likely to make continuous improvements in it [63], so employees have a more remarkable ability to create new ideas and implement them in their organization or service [20].

Therefore, employees who have high levels of work engagement are usually emotionally positive, physically and psychologically healthy, hard-working, dedicated and work-centred, generating the resources they need [64].

Thus, we can state that healthcare professionals have high levels of work engagement even though there are aspects of their work that cause stress, with work engagement tending to be inversely proportional to stress [10]. It was established in the study of [65] that concepts such as work engagement, well-being and job satisfaction impact nurses' health status, but few reliable scientific studies have revealed important information about work engagement in the healthcare sector [66].

A positive relationship between health professionals and the work environment will allow strengthening the bond between employees and professional practice, which provides for an improvement in the quality of services and care provided, which are work engagement and a positive mental state related to work [45,67] and, therefore, some specific characteristics of the work directly related to nurses' health, satisfaction and work engagement [13].

Although evidence demonstrates the essential consequences of work engagement on work motivation, the research of [68] shows that this is still insufficient and needs more cohesion. Therefore, for work engagement to have greater relevance nowadays, it will have to be seen as a unique mental model and different from the more traditional ones that have been used previously [69].

Given the above, we formulated the following research hypothesis:

**Hypothesis 4 (H4).** *Work engagement promotes nurses' innovative behaviour.*

### 2.5. Organizational Support

A theory of organizational support was suggested by [70], stating that when employees undertake administrative care, support and show affection, they perform a better job. Later on, the same author, together with other researchers [71], also mentioned that organizations caring for their employees improve their overall perception of the support offered by the organization.

Therefore, companies try to attract, keep and stimulate their employees through human resource management policies that offer incentives in the form of economic, financial, social or material rewards. Employees with good performance and commitment to the organization expect to receive resources that satisfy them at personal, family and professional levels [72].

In this study, we investigated the influence that organizational support has on nurses' innovative behaviour and work engagement, involving the Job Demands-Resources model and the theory of social exchanges, which is closely related to the discussed model [73].

Since the working conditions of the Job Demands-Resources model fall into two categories, job demands and resources, employees' well-being is thus attributed to job characteristics [22,70]. In this way, job demands are related to physical, social, psychological or organizational aspects that require effort from the employee, either physical or psychological [22]. On the other hand, job resources are related to physical, psychological, social or organizational characteristics, which allow employees to achieve their work goals, reduce demands and stimulate personal growth, learning and development [22].

Thereby, it can be said that the first assumption of the Job Demands-Resources model is that work resources, such as support from colleagues and supervisors, performance feedback and autonomy, contributes to a motivational process that leads to employees having work engagement and consequently improving their performance [11]. The second assumption is that work resources become more relevant and motivating when employees face higher demands in their workplace, such as workload and emotional/ mental demands [11], as happens in nurses' daily lives, especially in the pandemic that we are currently experiencing. Therefore, the model assumes the perspective of the positive organizational psychology movement, in which the employee's well-being is the result of the ideal balance between positive and negative aspects [74].

Based on the assumption that employees choose their degree of involvement at work as a response to the resources received from the organization, [73,75], they have used Social Exchange Theory to explain work engagement, as one of its principles states that relationships tend to evolve to integrate trust, loyalty and mutual commitment, provided that the parties involved respecting the rules of reciprocity. Similarly, [76] referred to individuals who relate to each other to maximize resource gains (material or emotional).

Accordingly, we formulated the following hypotheses:

**Hypothesis 5 (H5).** *Organizational support influences nurses' innovative behaviour.*

**Hypothesis 6 (H6).** *Organizational support influences nurses' work engagement.*

### 2.6. Job Demands-Resources Model

The application of the JD-R theory in this research aimed to support the relationships between the different dimensions under study since this model addresses the demands and resources to which professionals are subjected in their daily lives. In this sense, it is necessary to address the SDT theory briefly, the B&B theory and the SET theory that fit the JD-R model and the conceptual model of our research, based on the literature review conducted by [39] and the studies addressed by these authors, taking into account the demands and the individual and organizational resources addressed here.

Concerning the individual level, we can include the SDT and B&B theories to explain the relationships between stress, anxiety, work engagement, and nurses' innovative behaviour.

Thus, SDT aims to explain how work characteristics and individuals' characteristics influence their work engagement and behaviour, since, according to this theory, a determined employee is autonomous and has control over their work, which increases their commitment and innovative behaviour, without being assigned an external psychological burden [57].According to this theory, even if nurses face situations of stress and anxiety, provided that they are determined, autonomous and feel control over their work, they can use this stress and anxiety to their advantage through the development of innovative behaviours in the organizations where they work.

The B&B theory addresses, in turn, the need for an individual to be positive and to be involved in positive emotions that reinforce their personal resources as a way to experience innovative and lasting situations [77]. Thus, according to this theory, positive emotions, such as work engagement, help build personal resources because positive experiences and feelings lead individuals to have liberating, autonomous, creative and innovative behaviours [78,79]. Therefore, it may be said that high levels of work engagement will

contribute to high levels of innovative behaviours at work, and, in the case of nurses, they may use the positive effects of work engagement to build and manage creative and original solutions to the problems encountered in their daily work life [80].

Concerning organizational issues, SET (social exchange theory) states that when organizations offer resources and benefits to their employees, such as autonomy, they become more involved in their work by exchanging their involvement for benefits [75,81]. If these benefits are of an economic or social nature, employees feel the need to have attitudes of commitment that match the organizations expectations where they work [39]. Thus, this theory also explains the relationship between the organizational support offered and innovative behaviour, in that when expectations and demands of the job increase, employees tend to risk more and seek new and innovative alternatives to meet the needs of their work, provided there are signs that their organization will give them the necessary support and corresponding benefits, basing the employee's commitment on the interchange of the organization [28,39,75]. Healthcare organizations can provide more significant support to their employees, and nurses will inherently show more innovative behaviours. Therefore, organizations should identify organizational and individual factors that improve the relationships between nurses and supervisors [82].

### 2.7. Conceptual Model

Taking into account the literature review, the general research objective and the research hypotheses duly substantiated, we propose the following conceptual research model in Figure 1.

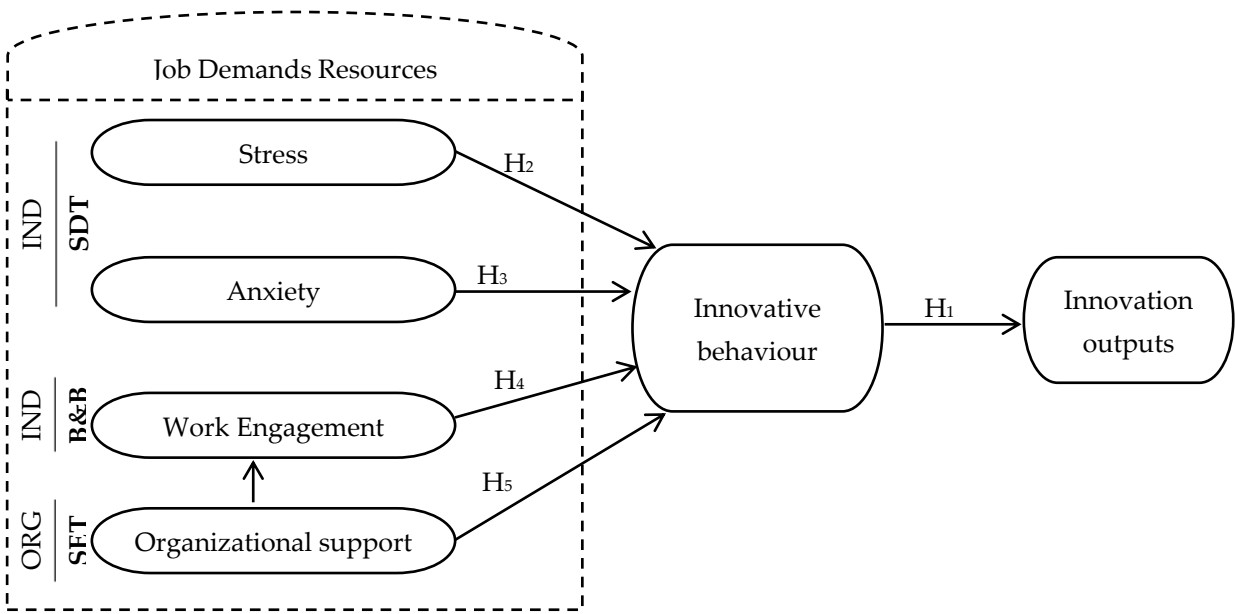

**Figure 1.** Conceptual research model.

### 3. Methodology

The study method of this research is classified as quantitative and, in terms of objectives, it is an exploratory and descriptive research.

The sample under study was composed of 738 nurses working in healthcare organizations in Portugal between 16 May and 5 October 2020. The sample was 81.6% female, and the most represented age group was between 36 and 45 years old with 38.2%. The majority of respondents were married or cohabiting (72.2%) and had a degree (40.8%) or a speciality/post-graduate degree (38.5%) as their academic qualifications.

The predominant professional category was that of Nurse (64.1%) followed by Nurse Specialist (33.6%) and, 76.7% of the respondents exercised their professional activity in

an organization that provides hospital care, in 55.6% of cases in the same hospital centre. Almost all professionals had a permanent contract with the organization (88.8%), with the most representative group being those with between 11 and 15 years of professional experience (20.2%). It should be noted that only 9.5% of the sample members held management/coordination positions.

### 3.1. Data and Sources

To empirically assess the model in Figure 1, a questionnaire with 65 questions divided into six parts was administered to 738 nurses working in healthcare units in Portugal. The variables were operationalized through scales already validated in previous studies. Lukeš and Stephan's (2017) scale was adapted to measure 'Innovative Behaviour' (20 items divided into six dimensions) and 'Innovation outputs' (3 items) [20]; Lukeš and Stephan's (2017), Kuratko, Hornsby, and Covin's (2014) scale for 'Organizational Support' (6 items) [20,83]; Schaufeli, Bakker and Salanova's (2006) UWES-9 (Utrecht Work Engagement Scale) was adapted to measure 'Work Engagement' [58]; Amirkhan' (2018) Short Stress Overload Scale (SOS-S) was used to measure stress (10 items) [84]; Spitzer, Kroenke, Williams and Lowe' (2006) Generalized Anxiety Disorder Scale (GAD-7), adapted to measure anxiety [85]. Seven-point Likert scales were used in the various measurement instruments.

The questionnaire was developed and tested in early 2020, then proceeded to actual data collection and implementation between 16 May and 5 October 2020.

### 3.2. Methods and Techniques of Analysis

To estimate the proposed model, structural equation modelling PLS (PLS-SEM) was applied, using the SmartPLS 3.0 software [86]. Among the reasons for using PLS-SEM are the low requirements regarding the underlying data distribution and sample size, compared to the covariance matrix-based structural equation model (CB-SEM), which has restrictions on distribution properties (multivariate normality), sample size, model complexity, factor identification and indeterminacy [87].

## 4. Presentation and Analysis of Results

### 4.1. Evaluation of the Structural Equation Model

In a first stage, the psychometric properties of the 2nd order latent variables of 'Innovative Behaviour' and 'Engagement' were assessed taking into account the recommendations mentioned by [88], assessing the constructs' reliability as well as their factor validity, convergent validity and discriminant validity, and subsequently determining the scores of the subconstructs of these constructs, which were subsequently used as manifest variables in the proposed final model [89]. The model was developed in two stages, comprising two sub-models. The first stage consisted of testing the measurement model (outer model), which defines how the hypothetical constructs or latent variables are operationalized by the observed or manifest variables. The second step consisted of testing the structural model (inner model), which defines the causal relationships among the latent variables. The regression coefficient paths with confidence levels greater than 95% ($p < 0.05$) were considered significant.

### 4.2. Measurement model (Outer Model)

Reliability, factor, convergent, and discriminant validity were assessed to determine the proposed measurement model. As shown in Table 1, the latent variable 'Anxiety' was composed of seven manifested variables, 'Stress' of 10, 'Work Engagement' of three, 'Organizational Support' of six, 'Innovative Behaviour' of six and, finally, 'Innovation outputs' of three.

**Table 1.** Final model, reliability and validity of the constructs.

| Items | λ | CR | α Cronb. | AVE |
|---|---|---|---|---|
| Anxiety (over the last 2 weeks, how often have you been bothered by the following problems) | | | | |
| Ans_1-Feeling nervous, anxious or on edge | 0.886 *** | | | |
| Ans_2-Not being able to stop or control worrying | 0.917 *** | | | |
| Ans_3-Worrying too much about different things | 0.905 *** | | | |
| Ans_4-Trouble relaxing | 0.922 *** | 0.964 | 0.958 | 0.793 |
| Ans_5-Being so restless that it is hard to sit still | 0.908 *** | | | |
| Ans_6-Becoming easily annoyed or irritable | 0.887 *** | | | |
| Ans_7-Feeling afraid as if something awful might happen | 0.805 *** | | | |
| Stress | | | | |
| Stress_1-In the past week, have you felt inadequate? | 0.704 *** | | | |
| Stress_2-In the past week, have you felt swamped by your responsibilities? | 0.729 *** | | | |
| Stress_3-In the past week, have you felt that the odds were against you? | 0.778 *** | | | |
| Stress_4-In the past week, have you felt that there wasn't enough time to get to everything? | 0.805 *** | | | |
| Stress_5-In the past week, have you felt like nothing was going right? | 0.746 *** | | | |
| Stress_6-In the past week, have you felt like you were rushed? | 0.793 *** | 0.934 | 0.929 | 0.588 |
| Stress_7-In the past week, have you felt like there was no escape? | 0.809 *** | | | |
| Stress_8-In the past week, have you felt like things kept piling up? | 0.834 *** | | | |
| Stress_9-In the past week, have you felt like just giving up? | 0.660 *** | | | |
| Stress_10-In the past week, have you felt like you were carrying a heavy load? | 0.796 *** | | | |
| Work Engagement | | | | |
| WE_A–Absorption | 0.862 *** | | | |
| WE_D–Dedication | 0.939 *** | 0.932 | 0.889 | 0.820 |
| WE_V–Vigour | 0.913 *** | | | |

**Table 1.** *Cont.*

| Items | λ | CR | α Cronb. | AVE |
|---|---|---|---|---|
| Organizational Support | | | | |
| OrgSup_1-The way of rewarding in my organization motivates employees to come up with new ideas and procedures | 0.889 *** | | | |
| OrgSup_2-My organization encourages employees who have innovative ideas | 0.935 *** | | | |
| OrgSup_3-My organization has ensured sufficient resources to support the implementation of new ideas | 0.936 *** | | | |
| OrgSup_4-My organization provides time for employees to put their ideas and innovations into practice | 0.935 *** | 0.974 | 0.968 | 0.862 |
| OrgSup_5-My organization often recognizes employees who take individual risks for their willingness to defend new projects, whether they are successful or not | 0.953 *** | | | |
| OrgSup_6-My organization encourages employees to talk to colleagues in other departments about ideas for new projects | 0.921 *** | | | |
| Innovative Behaviour | | | | |
| IDEAGEN-Generation of ideas | 0.820 *** | | | |
| IDEASEA–Search for ideas | 0.802 *** | | | |
| IDEACOM–Communication of ideas | 0.909 *** | 0.945 | 0.930 | 0.741 |
| IMPL–Start of implementation | 0.880 *** | | | |
| INVOL–Involvement of colleagues | 0.879 *** | | | |
| OVERC–Overcoming obstacles | 0.871 *** | | | |
| Innovation *Outputs* | | | | |
| OUT_01-I am often successful at work when I put my ideas into practice | 0.887 *** | | | |
| OUT_02-Many things created by me are used in our organization | 0.879 *** | 0.919 | 0.868 | 0.791 |
| OUT_03-I have always implemented improvements in the places where I worked | 0.902 *** | | | |

*** $p < 0.001$.

The reliability of the six constructs that make up the model was assessed through the analysis of the composite reliability values of Jöreskog [90] and Cronbach's $\alpha$, which were higher than 0.7, with minimum HR and Cronbach's $\alpha$ values of 0.919 and 0.868, respectively, in the 'Innovation outputs', thus ensuring the reliability of the constructs (Table 1). Factor validity, on the other hand, was assessed through the analysis of the factor loadings of the indicators ($\lambda$), which, the except for item "Stress_9-In the past week, have you felt like just giving up?" ($\lambda = 0.660$) of the 'Stress' construct, were higher than 0.708, thus confirming the factor validity. It should be noted that we chose to keep this item for theoretical reasons. Because it did not influence the construct's reliability and convergent validity. Convergent validity was assessed by determining the value of the average variance extracted (AVE), whose value was higher than 0.588 (>0.50) in all cases and was, therefore, ensured [91].

Discriminant validity was assessed by the heterotrait–monotrait (HTMT) ratios of correlations criterion [92]. As shown in Table 2, all correlation ratios were below the threshold value of 0.900 with only one ratio slightly above the conservative threshold of 0.850, between 'Innovative behaviour' and 'Innovation outputs' (0.861), so there is discriminant validity.

**Table 2.** Heterotrait–monotrait (HTMT) ratio.

| | (1) | (2) | (3) | (4) | (5) | (6) |
|---|---|---|---|---|---|---|
| (1) Anxiety | | | | | | |
| (2) Stress | 0.773 | | | | | |
| (3) Work Engagement | 0.246 | 0.228 | | | | |
| (4) Organizational Support | 0.057 | 0.057 | 0.420 | | | |
| (5) Innovative Behaviour | 0.107 | 0.147 | 0.524 | 0.273 | | |
| (6) Innovation Outputs | 0.112 | 0.153 | 0.458 | 0.341 | 0.861 | |

### 4.3. Structural Model (Inner Model)

An approximate descriptive index of general adjustment implemented in structural equation modelling using PLS is the mean square of standardized residuals (SRMR). A value of 0 indicates a perfect fit, although fit values below 0.08 reveal an adequate fit. The model proposed in this study has an SRMR value of 0.068 (<0.08), which has an adequate adjustment.

Note that the evaluation of the structural model and its predictive capacity is carried out by the R2 of the latent endogenous variables [93], but also by the size of the f2 effects [94]. As can be inferred from Table 3, the R2 value for 'Innovative behaviour' was 0.280, that of 'Work engagement' 0.152 and that of 'Innovation outputs' 0.610, so all were higher than the acceptable cutoff point 0.1 [95]. The effect size (f2) complements R2 and considers the relative impact of a particular exogenous variable on an endogenous variable through changes in R2 [94]. Reference [94] suggested f2 values of 0.02, 0.15 and 0.35 for small, medium and large predictive variable effects. For the model under analysis, it is possible to see through Table 3 that the largest effects occur between 'Innovative behaviour' and 'Innovation outputs' with an f2 value of 1.562, verifying average effects between 'Work engagement' and 'Innovative behaviour' (0.266) and between 'Organizational support' and 'Work engagement' (0.179).

Similarly, the predictive relevance of the model was assessed using the Stone-Geisser Q2 statistic. This procedure was performed following the blindfolding approach (considering 7 the default distance), thus examining the model's predictive power [96]. It was found that the Q2 value ranges from 0.123 in the construct 'Work engagement' to 0.476 in the construct 'Innovation outputs', which is why it is higher than zero in all constructs, thus suggesting the predictive relevance of the model [93].

**Table 3.** Size of the effects of predictor variables on endogenous variables.

| Path | $R^2$ | $f^2$ | $f^2$ Effect |
|---|---|---|---|
| Innovative Behaviour → Innovation outputs | 0.610 | 1.562 | Large |
| Stress → Innovative Behaviour | 0.290 | 0.014 | - |
| Anxiety → Innovative Behaviour | 0.290 | 0.011 | - |
| Work Engagement → Innovative Behaviour | 0.290 | 0.266 | Medium |
| Organizational Support → Innovative Behaviour | 0.290 | 0.008 | - |
| Organizational Support → Work Engagement | 0.152 | 0.179 | Medium |

Figure 2 refers to the SmartPLS output regarding the model with the control variables, with the R2 values represented within the latent endogenous variables, the regression coefficients of the structural model (inner model), as well as the representation of all items represented in the model with their respective factor loadings (outer model).

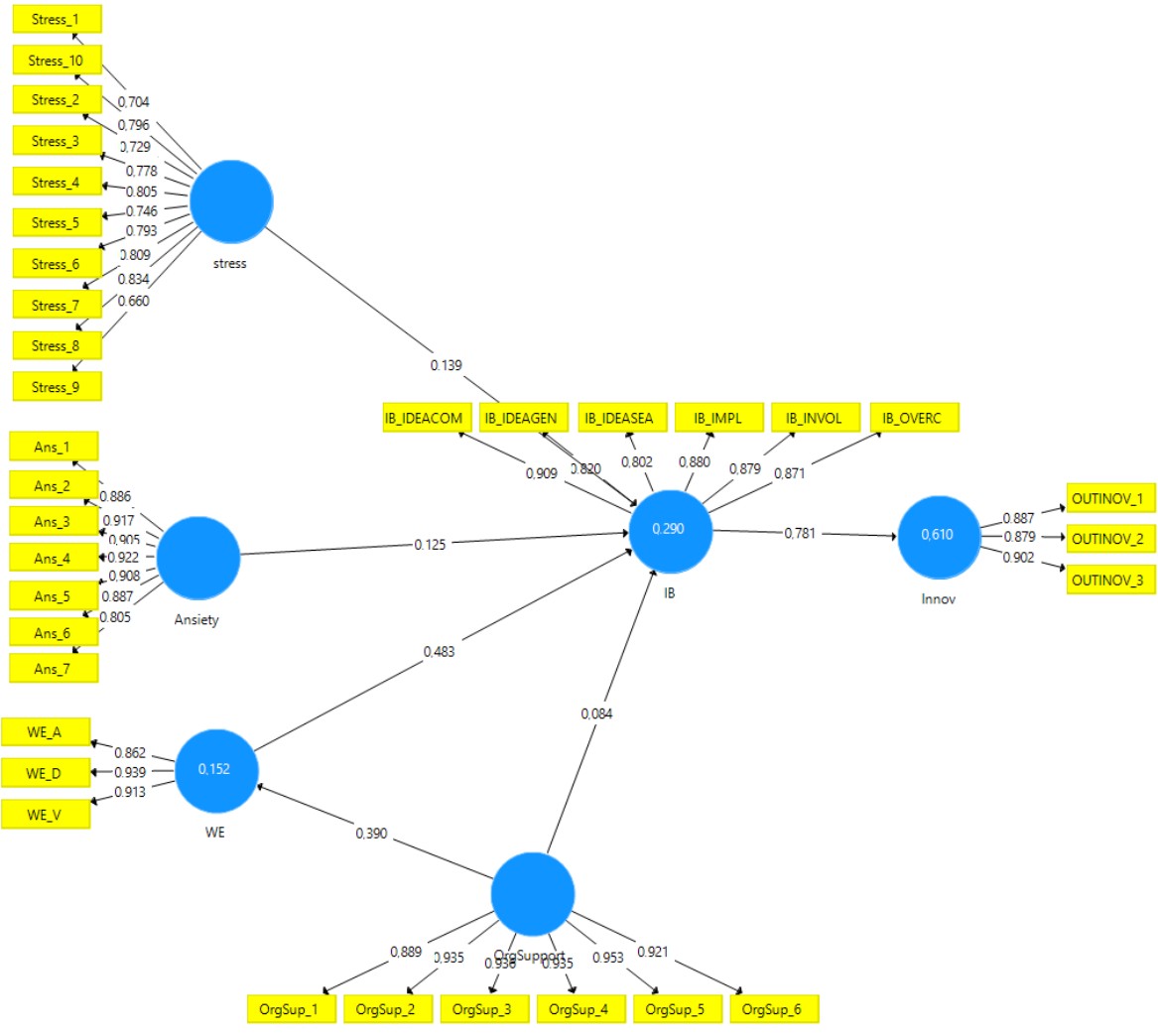

**Figure 2.** Output of the structural model in SmartPLS with indication of $R^2$ values, regression coefficients and factor loadings.

### 4.4. Hypotheses and Research Questions

After the structural equation modelling analysis, we illustrate in Figure 3 the structural model with the regression coefficients and statistical significance, and for a more detailed analysis, we summarize in Table 4 the results obtained regarding the formulated hypotheses.

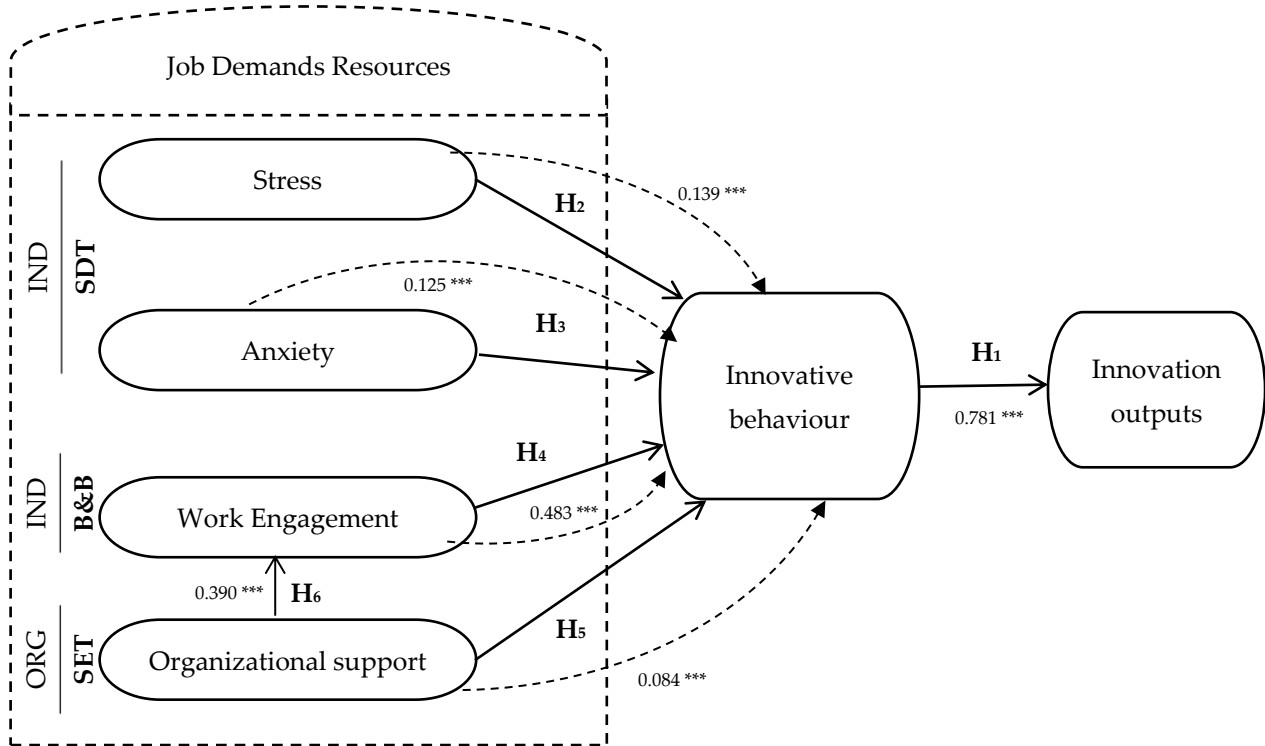

**Figure 3.** Structural model (regression coefficients and statistical significance). Note: *** $p \leq 0.001$.

**Table 4.** Hypotheses results.

| Path Direct Effect | Regression Coefficient (β) | *p* | Is the Hypothesis Supported? |
|---|---|---|---|
| H1: Innovative Behaviour → Innovation | 0.781 *** | <0.001 | Yes |
| H2: Stress → Innovative Behaviour | 0.139 ** | 0.002 | Yes |
| H3: Anxiety → Innovative Behaviour | 0.125 ** | 0.004 | Yes |
| H4: Work Engagement → Innovative Behaviour | 0.483 *** | <0.001 | Yes |
| H5: Organizational Support → Innovative Behaviour | 0.084 * | 0.014 | Yes |
| H6: Organizational Support → Work Engagement | 0.390 *** | <0.001 | Yes |

Note: *** $p < 0.001$; ** $0.001 \leq p < 0.010$; * $0.010 \leq p < 0.050$.

We found that the seven proposed hypotheses all have significance and are all supported, with H1—"Nurses' innovative behaviour promotes innovation outputs"—proving to be not only positive and significant (β = 0.781; $p < 0.001$) but also having the highest regression coefficient in the model, and H5—"Organizational support promotes nurses' innovative behaviour"—although also significant ($p = 0.014$), being the one with the lowest regression coefficient in the model (β = 0.084).

## 5. Discussion of Results

For [97], the Job Demands-Resources model integrates two basic psychological processes: a health-damaging process, such as stress, triggered by excessive work demands and lack of resources; and a motivational process, such as work engagement, pending high resources in the organization, which leads to positive outcomes such as organizational commitment, the intention to stay in the organization and high performance.

From the values presented in Figure 2, it is possible to evaluate the result of the research hypotheses formulated. As can be ascertained in the evaluation of the structural model, 'Innovative behaviour' has a significant impact on 'Innovation outputs' supporting

hypothesis 1. It should also be reinforced that this hypothesis is positive and relevant. It is also of the highest significance, demonstrating the positive influence that nurses' innovative behaviour has on innovation outputs.

Thus, with regard to the construct 'Innovative behaviour', we found that 'Communication of ideas' (0.909) is the dimension with the highest factorial loading and the item of the questionnaire that most influences it is "I try to show my colleagues the positive side of new ideas" (0.922). This is followed by the dimension 'Start of implementation' (0.880), with the item "To implement new ideas, I look for new technologies, processes and procedures" (0.925) and the dimension 'Involvement of colleagues' (0.879), with the item "I try to involve key decision-makers in the implementation of an idea" (0.926). 'Overcoming obstacles', 'Generation of ideas' and 'Search for ideas' have factor loadings of 0.871, 0.820 and 0.802, respectively.

Concerning the 'innovation outputs', the item with the highest loading is "I have always implemented improvements in the places where I worked" (0.902), followed by the item "I am often successful at work when I put my ideas into practice" with the loading of 0.887, and finally the item "Many things created by me are used in our organization" with 0.879, which shows how much nurses influence the daily lives of their teams and the organizations to which they belong.

Nurses' innovative behaviour can be seen as the stimulus given to professionals to use their acquired knowledge and skills to create and develop new ways of working, thus being able to use technologies, systems creatively, theories and interested partners to further enhance and evaluate clinical practice [29], which ultimately corroborates H1, since only with innovative behaviour positively influencing the innovation outputs is it possible to obtain more new innovative products/procedures, especially in the healthcare sector.

### 5.1. 'Stress' and 'Anxiety' Influence 'Innovative Behaviour' Supporting Hypotheses 2 and 3, Respectively

With regard to the dimension 'stress', the items that most influenced it to show that, in the week before answering the questionnaire, nurses felt: "like things kept piling up" (0.834), "like there was no escape" (0.809) and "there wasn't enough time to get to everything" (0.805).

Most of the previous studies do not corroborate the results we have obtained insofar as they state that stress does not positively influence innovative behaviour (e.g., ref. [1,98–100]. However, reference [28] refer that stress factors stimulate innovative behaviour as long as under specific and ideal conditions, which was also found in our study, and may be explained by the fact that we are studying nurses who, in themselves, are already usually innovative professionals even more so in times of the COVID-19 pandemic, in which it is necessary to innovate to overcome the difficulties experienced in the daily life of these professionals.

Concerning the dimension 'anxiety', the nurses were asked how often they had certain feelings in the 2 weeks before answering the questionnaire. They reported feeling: "bothered by trouble relaxing" (0.922) and "bothered by not being able to stop or control worrying" (0.917), which were the items with the highest loading in this dimension.

As anxiety is a reaction deeply related to fear, which in certain situations may be exaggerated or excessive, it can, however, also be a mild situation in certain circumstances, ultimately playing an adaptive role in human and innovative development [3]. Thus, in a positive relationship between feelings, communication and implementation efforts, it can be a crucial factor for innovative behaviour to happen [26], corroborating hypothesis 3 (anxiety influences innovative behaviour).

### 5.2. 'Work Engagement' Has a Significant Effect on 'Innovative Behaviour', Supporting Hypothesis 4

Thus, in terms of the construct work engagement, we found that the dimension 'Dedication' presents the highest factor loading (0.939), with the item with the highest loading being "My job inspires me" (0.958), followed by the dimension 'Vigor' with a factor loading of 0.913, with the item with the highest loading "At my work, I feel like I am

bursting with energy" (0.977) and the dimension 'Absorption' (0.862) and the individual item "I am immersed in my work" (0.908).

Previous studies have proven that work engagement is positively associated with innovative behaviour (e.g., [24,35,101]), therefore nurses' work engagement and innovative behaviour' are beneficial to improve the quality, efficiency and competitiveness of nursing services [101], which is corroborated by our results, showing that 'work engagement' influences nurses' 'innovative behaviour', insofar as the more engaged they are at work the more innovative behaviours they have, which is crucial in a pandemic.

It was found that 'Organizational Support' impacts 'Innovative Behaviour' and 'Work Engagement', thus supporting Hypotheses 5 and 6, respectively.

Performing a more careful analysis of the dimensions of 'Organizational support' we find that the dimensions with the highest factorial loadings are: "My organization often recognizes employees who take individual risks for their willingness to champion new projects, whether they are successful or not" (0.953); "My organization has ensured sufficient resources to support the implementation of new ideas" (0.936); "My organization encourages employees who have innovative ideas" (0.935); "My organization gives employees time to put their ideas and innovations into practice" (0.935).

Changes happen quickly in organizations and in the business world in general, which leads institutions to suffer pressure to respond positively and in a timely and effective way, which makes it necessary for the leader to provide more significant support to employees so that they engage in the work in a committed way [33]. Therefore, it is essential in several organizations to improve employees' innovative behaviour, which represents a significant challenge for management [102,103]. Leaders and organizations can, through organizational support, provide support or resources to facilitate employees' attempts to bring to work positive change, new ideas or innovative behaviour [27], thus corroborating the results of our study, which shows that organizational support positively influences innovative behaviour.

Similarly, and to the extent that employees are more likely to exchange their commitment for resources and benefits provided by their organization [75], it is assumed that if an organization offers employees economic and socioeconomic resources, employees feel obliged to respond with a corresponding level of commitment [73,104]. Conversely, they may not be engaged if their organization fails to deliver on its promise by preceding adequate compensation, promotion, job security, training opportunities, and other desirable incentives [73,104]. Therefore, it is understandable that our research also concludes that organizational support impacts on work engagement.

## 6. Conclusions and Future Recommendations

### 6.1. Conclusions

The Job Demands-Resources model was included in this research, because this model was related to working conditions, in which work demands and resources were included, according to [22]. As time passed and with the various revisions of the model, it was possible to add the motivational role related to high resources [8], positive leadership [23] and more recently, the leader's feedback [24], and these changes are now also mainly related to the organizational support and management support addressed by [20] when studying innovation outputs, thus encompassing the constructs we are investigating.

It was stated by [105] that due to the increase in the average life expectancy and the incidence of older people with dementia, the decrease in the birth rate, the increasing need for long-term care with diminished resources, innovation would be necessary to overcome these types of challenge. Thus, according to [41], receiving adequate knowledge, an orientation towards change and adaptation to the external environment increase the organizational responsiveness to needs and a collective concern for the continuous improvement of the organization itself, thereby promoting the innovative behaviour of healthcare professionals and, indirectly, the innovation outputs.

Ability and opportunity relate directly to employees' innovative behaviour [106], and employees with such behaviours generate original and valuable ideas and apply those ideas to practice [107]. Therefore, if the person/organization relationship is adjusted, it will positively affect the employee's innovative behaviour, but this will only happen if the person is psychologically empowered and given the independence to carry out their activities at work [108]. It was, therefore, necessary to understand the relationship between a profession such as nursing, stress, anxiety and the innovative behaviour of these professionals.

According to [100], the most stressful type of work values excessive demands and pressures that do not match workers' knowledge and abilities, where there are few opportunities to exercise any choice or control, and where there is little support from others. Workers are less likely to experience work-related stress when the demands and pressures of the job are matched to their knowledge and abilities, and control can be exercised over their work and how they do it [100]. However, there are situations in the nursing profession for which nurses and other healthcare professionals may not be prepared or equipped with knowledge or resources, as was the case of the COVID-19 pandemic, which may generate stressful situations.

According to our results, for nurses to be able to overcome the difficulties they encounter every day in the field, they must have innovative behaviours and work engagement, as well as support from the organizations.

When employees show work engagement, they make more contributions to the organization, generate more ideas and produce initiatives that impact innovation [109]. Similarly, [63] states that work engagement is related to innovative behaviour through employee performance. If employees feel interested and enthusiastic at work, they are likely to bring continuous improvements. Therefore, for [110], when there is work engagement, employees feel indirectly more stimulated to be creative, showing a more open mind, stimulating the organizations to innovate, which will make them more active and responsible.

According to the reciprocity rule of the social exchange theory, organizational support encourages employees to work hard to give back to the organization; thus, organizational support is influential in significantly increasing employee performance since the relationship between the employee and their organization involves mutual interdependence [111]. Conversely, if employees are not rewarded for their efforts because the organization does not recognize the action or because it does not have the necessary resources to reward them, employees may decrease their efforts and become disinterested in their work [112], leading nurses, in this specific case, not to present innovative behaviours, nor demonstrate the necessary work engagement to face COVID-19.

When a disease appears in peoples' lives, they all take different attitudes depending on their beliefs, values, and personal history. These attitudes may be more or less correct depending on each persons' final objective and the point they want to reach. However, when a pandemic occurs, everything takes on gigantic proportions. In reality, this reveals the most intuitive and creative side of each person because everyone has the purpose of staying alive and healthy. In the case of nurses, this need is even more remarkable because, in addition to having this desire for themselves and their family and friends, they have the same passion for people in general. Associated with this idea, they also feel the need to fulfil their duty in the best possible way, using innovative ideas to do so, regardless of the stress, anxiety and support received by the organisation where they work. That is why it is necessary to look at nurses more and more as extraordinary and essential professionals during health and illness, and of course, during a pandemic. These are the health professionals who prove to be innovative even in the greatest adversity, and as such, they can and should be a factor for top research and support so that they can be helped and at the same time help others.

*6.2. Theoretical Contributions*

The theme addressed in this study contributes to the scientific advancement in the sense that innovative behaviour and innovation outputs are increasingly relevant to broaden scholars' knowledge that will improve organizations, especially in healthcare organizations.

Moreover, the influence of stress, anxiety, work engagement and organizational support on nurses' innovative behaviour during the COVID-19 pandemic allows us to understand that these matters should be increasingly deepened and studied to grow as a profession, contributing to an improvement in patient's quality of life.

Finally, it is also necessary to emphasize the importance of the Job Demands-Resources model approach in this research and its application in theoretical terms to healthcare.

*6.3. Practical Contributions*

With this study, we contributed to more excellent knowledge about nurses' innovative behaviour and innovation outputs and the dimensions that influence it, which is undoubtedly advantageous knowledge for healthcare managers.

Thus, the results obtained can increase and improve nurses' and other healthcare professionals training, to increase the innovative behaviour that generates the innovation outputs, allowing for faster and more effective discovery of valuable innovations in this and the next pandemic. In addition, the leaders and managers closest to the professionals must understand the importance of work engagement for this same innovation to happen.

Another contribution that our research can make comes from the initial training of nurses insofar as they can and should be encouraged to display innovative behaviours from an early age so that they have a team spirit and learning from initial training and then be able to understand the importance of these attitudes when talking about a pandemic period.

*6.4. Implications and Limitations of the Research*

The condition of being a nurse is in itself an implication in this research because these professionals are put to the test in their daily lives, with frequent adverse situations, in which uncertainty, grief and the risk of contracting diseases that they may transmit to their relatives, is added to personal difficulties such as stress and anxiety they experience because of often working shifts, having to manage their emotional and personal side depending on the support given by the organization. These issues are advanced by the Job Demands-Resources model widely studied in this research.

Stress and anxiety influence the nurses' innovative behaviour to the extent that we are in the COVID-19 period. There was a need for these professionals to adapt and readapt regardless of their fears about such an unknown disease. To this end, we should bear in mind that data were collected between May and October 2020, which was in the middle of the first wave of COVID-19, may have become a limitation because the nurses were affected and exhausted by facing the unknown and its consequences.

It should also be mentioned that organizational support and work engagement positively influenced innovative behaviour, which influenced innovation outputs, stimulating research and rapid investigation as a way of facing the pandemic. Despite this, it can be seen that there is a difficulty on the part of top managers and those responsible for the healthcare area in Portugal in observing the results of studies carried out and taking action in response to them.

*6.5. Future Recommendations*

In the future, we believe it would be interesting to investigate the nursing management area further to understand their point of view regarding their innovative behaviour and the personal and organizational factors that influence it.

Other studies that could be carried out would be: doing the same research with other healthcare professionals; repeating the study in a phase-out of the COVID-19 pandemic to verify the differences in the results obtained; or else sectioning the questionnaires by services with a particular interest in urgency/emergency/intensive care as a way of

understanding whether nurses' innovation in these areas is higher, lower or equal compared to nurses in general services.

We could also expand the study by embracing the area of quality to reinforce the importance of innovation in providing nursing care.

**Author Contributions:** Conceptualization, A.M.C.; methodology, A.M.C.; validation, C.S.M. and G.S.; investigation, A.M.C., C.S.M. and G.S.; resources, G.S.; software, G.S.; data curation, A.M.C.; writing—original draft preparation, A.M.C.; writing—review and editing, A.M.C.; visualization, A.M.C.; supervision, C.S.M., G.S.; funding acquisition, C.S.M. and G.S. All authors have read and agreed to the published version of the manuscript.

**Funding:** The paper was funded by national funds, through the FCT—Portuguese Foundation for Science and Technology under the project UIDB/04011/2020.

**Institutional Review Board Statement:** Not applicable.

**Informed Consent Statement:** Informed consent was obtained from all subjects involved in the study.

**Data Availability Statement:** Not applicable.

**Conflicts of Interest:** The authors declare no conflict of interest.

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
