# Peer review of "Organizational and Personal Factors That Boost Innovation: The Case of Nurses during COVID-19 Pandemic Based on Job Demands-Resources Model"

_sustainability, doi:10.3390/su14010458_

Round 1

Reviewer 1 Report

The research paper's abstract does not provide a bird's eye view (snapshot view) of what is being discussed throughout the paper. -The linkage between the manuscript's objective and why the research was necessary was not well established?After reading the authors ' abstract cited, the reader is likely to be clueless and confused about the contents.

- This paper is interesting, and it deals with a topic central to the field of research.  The review of the literature is satisfactory, and the hypothesis is sustainable and well presented. The framework of the analysis is clear, as well as the methodology. Also, it is well written.

Organizational Collaborative Culture as a Source of Managing Innovation, World Applied Sciences Journal, 24 (5):582–587. ISSN: 1818–4952

- I suggest rewriting the introduction part and critically analyze the literature.I would also suggest to the authors extend their literature review regarding the field. It is not clear how this paper contributes to the extant literature.

- The style of introduction must be coherent, and it should explain what the problem is, what has been researched in previous academic literature in this area, and what actually gap exists .further, how this study fulfills this research gap

Effects of the buyer-supplier relationship on social performance improvement and innovation performance improvement, International. Journal of Applied Management Science, Vol. 11, No. 1.pp.  21–35

"Environmental innovation strategy and organizational performance: Enabling and controlling uses of management control systems." Journal of Business Ethics 151, no. 4 (2018): 1139-1160.

-The authors need to address the limitations of their chosen methodology more clearly in terms of recall or response bias.

- The manuscript is potentially original contributive but needs a significant rewriting

- How the results of this study can be generalized to other companies.

Author Response

REVIEWER #1

We appreciate your time and effort in reviewing our paper and greatly value your constructive comments. We respond to your remarks below. Thank you.

The research paper's abstract does not provide a bird's eye view (snapshot view) of what is being discussed throughout the paper. -The linkage between the manuscript's objective and why the research was necessary was not well established? After reading the authors ' abstract cited, the reader is likely to be clueless and confused about the contents.”

R: Thanks for this suggestion. We have improved and rewrite the abstract section to relate the purpose of the study and the need for the research undertaken (see p. 1, text in blue).

I suggest rewriting the introduction part and critically analyze the literature. I would also suggest to the authors extend their literature review regarding the field. It is not clear how this paper contributes to the extant literature.

The style of introduction must be coherent, and it should explain what the problem is, what has been researched in previous academic literature in this area, and what actually gap exists, further, how this study fulfills this research gap.”

The authors need to address the limitations of their chosen methodology more clearly in terms of recall or response bias.

R: Thanks for this suggestion, however, in the authors' perspective, there is no clear bias in the methodology chosen, neither in the collection of responses nor in the responses obtained. During Covid-19 in which the transfer of physical material (such as questionnaires) was not allowed due to the risk of transmission of the disease itself, the only hypothesis that seemed possible to us was even the dissipation of the questionnaire via the internet. In the same way, and since nurses were busier with their work than usual, it seemed to us that dissipating the questionnaire through personal and work networks would be the easiest way to obtain more answers. As such, and inherently due to the moment nurses are experiencing, it did not seem to us that there was a bias in their answers. On the contrary, it seemed that their answers are more sincere and objective, due to the situations they are experiencing in their daily lives.

How the results of this study can be generalized to other companies.”

R: Thanks for the suggestion. However, the authors believe that, in reality, this study cannot be generalized to other types of companies because each company operates in a specific way according to its typology, and healthcare organizations belong to a very particular type of company. In our perspective, the nurses' point of view and behaviour cannot be generalized in the same way as that of other workers outside the healthcare sector, particularly because there is always the possibility that, even within the healthcare sector, other professionals have different innovative behaviour than nurses in a pandemic moment. Thus, we believe that, although the questionnaire can, in part, be applied to other professionals from other areas, we cannot generalize the results to other companies.

Thank you for your helpful comments and suggestions which have contributed to the improvement of this paper.

Reviewer 2 Report

It is pleasant to find an article in this sense as part of the academy, as the authors indicate, it helps to strengthen some hypotheses so that these types of organizations make decisions in this regard.

As for the abstract, I suggest improving the wording that adds value and reflects the importance of the content, in the first instance it becomes very general and the power of attraction is not so high for the reader to continue reading.

Regarding the literature and references, interesting sources and similar studies are observed, which strengthens the communication proposal, however, by mentioning in the title the Covid-19, the references of recent years and that they contextualize about the new forms of work and as people develop in them and the relevance of the value they have taken, especially in the health sector, are not enough, therefore the recent literature must be strengthened, which if it is clear, is that the work commitment, the behavior and related aspects between people and organizations have changed. Get more recent sources of information since the percentage of the last 5 years is low.

Regarding the population, there is a case that caught my attention and that is that it refers to Portugal as a country, but it indicates that 55.6% of the people work in the same hospital. In this sense, it is necessary to clarify this so that the reader understands that the case refers mostly to a single organization and that to generalize it could be strengthened with a new contribution or replica, take into account other people from different institutions and be able to compare the validity of the hypotheses and reach country situations.

This would result in valuable information for public policies regarding the management of personnel, especially in the health sector, which has particularities regarding behavior and work commitment, and which are sometimes influenced by issues of salary competitiveness.

The results in scientific terms are well expressed, but it is relevant that science appropriates effective communication for those who really should take advantage of this product. Communicating science should not obey only to capture statistical data and compare in the discussion if it is the same or different from other institutions, how to achieve that the writing achieves an impact on the reading of the senior management of medical centers, on those responsible for the area of human management, so that they can implement actions that result in organizational benefit.
Although the conclusions are relevant, a better effort could be made and to be able to communicate better not only in an academic writing, but also in a writing that manages to impact the organizations of the health sector not only in Portugal but also in other countries that have similar problems.

In terms of similarity, ithenticate has reported a percentage higher than 30% that for this case should not be allowed, although it is true they are not extensive situations, but the invitation is necessary for the team to improve the writing and avoid these situations.

Author Response

REVIEWER #2

We appreciate your time and effort in reviewing our paper and greatly value your constructive comments. We respond to your remarks below. Thank you.

As for the abstract, I suggest improving the wording that adds value and reflects the importance of the content, in the first instance it becomes very general and the power of attraction is not so high for the reader to continue reading.

R: Thanks for this suggestion. We have improved and rewrite the abstract section to demonstrate the importance of the study and the value that the results obtained add to science, making it more attractive to read (see p. 1, text in blue).

“(…) by mentioning in the title the Covid-19, the references of recent years and that they contextualize about the new forms of work and as people develop in them and the relevance of the value they have taken, especially in the health sector, are not enough, therefore the recent literature must be strengthened, which if it is clear, is that the work commitment, the behavior and related aspects between people and organizations have changed. Get more recent sources of information since the percentage of the last 5 years is low.”

Regarding the population, there is a case that caught my attention and that is that it refers to Portugal as a country, but it indicates that 55.6% of the people work in the same hospital. In this sense, it is necessary to clarify this so that the reader understands that the case refers mostly to a single organization and that to generalize it could be strengthened with a new contribution or replica, take into account other people from different institutions and be able to compare the validity of the hypotheses and reach country situations.”

The results in scientific terms are well expressed, but it is relevant that science appropriates effective communication for those who really should take advantage of this product. Communicating science should not obey only to capture statistical data and compare in the discussion if it is the same or different from other institutions, how to achieve that the writing achieves an impact on the reading of the senior management of medical centers, on those responsible for the area of human management, so that they can implement actions that result in organizational benefit.”

R: Thanks for this suggestion, however, the authors believe that the communication of the results expressed in this article is well expressed both at a scientific level and at the level of the professionals who may benefit from them. The way in which the ideas to be retained from this article are expressed is, in the authors' opinion, easy to understand for other researchers, health care managers and even people in the area of human resources, as well as nurses themselves. We believe that any of these professionals will be able to understand what is explicit in the results and use them to improve their organization.

Although the conclusions are relevant, a better effort could be made and to be able to communicate better not only in an academic writing, but also in a writing that manages to impact the organizations of the health sector not only in Portugal but also in other countries that have similar problems.”

R: Thanks for this suggestion, although the authors agree that this study can be used as a basis for other international studies, we do not believe that the major objective is to create an impact on organizations in other countries because each country is unique, its nurses are unique and the way their institutions deal with Covid-19 is also unique. Therefore, the authors believe that it will be difficult to transport the results obtained from one country to another, as the healthcare policies themselves are different, not to mention the inherent culture of the healthcare professionals themselves.

In terms of similarity, ithenticate has reported a percentage higher than 30% that for this case should not be allowed, although it is true they are not extensive situations, but the invitation is necessary for the team to improve the writing and avoid these situations.”

R: Thank you for the suggestion, but regarding this point, the authors have reviewed the article and in fact, the similarities found to refer to the practical part of the work, which makes this change difficult. They are objective sentences and difficult to restructure. In addition, as mentioned, the situations are not extensive, but since several dimensions are related, the possibility of this part being somewhat replicated is not totally impossible.

Thank you for your helpful comments and suggestions which have contributed to the improvement of this paper.

Round 2

Reviewer 1 Report

- I appreciate the novelty of the author's contributions, but still, authors need to grasp deep ideas on developing the research gap in the Introduction. The manuscript concerning the presentation of ideas shows improvement, but there are still some sentence structure mistakes and develop more clear research gap with the support of previous published studied in the domain of your study aim. However, to genuinely contribute with an article that is expected to synthesize previous contributions in the field and point our further research, i deem that the authors need to do some more work, particularly to sharpen the current manuscript.

-The last paragraph of the Introduction must be done a summary (resume) of the paper, i.e., a clear idea about what will be studied in the paper. I could not identify this. The last paragraph's purpose in the Introduction section is to summarize the main points, restate the paper's main idea, and show how the paper statements were proven.

-A brief literature review or a brief theoretical background section is required to define the study's main concept and explain. The current conceptual model does not explain well which theory the author applied. Thus, I suggest that this study   Considering that so much research has already been done within these areas, the paper needs a stronger and more focused approach and a more precise positioning. If so, it is essential to discuss the state of knowledge within these areas in much more detail, previous insights and results that this study may relate to, definitions of the main concept

- Implications for future research may also be included in the conclusion at the end. This research has article has created a lively discussion on so many issues that were hitherto unheard of and not addressed.

Author Response

We appreciate your time and effort in reviewing our paper and greatly value your constructive comments. We respond to your remarks below. Thank you.

“I appreciate the novelty of the author's contributions, but still, authors need to grasp deep ideas on developing the research gap in the Introduction. The manuscript concerning the presentation of ideas shows improvement, but there are still some sentence structure mistakes and develop more clear research gap with the support of previous published studied in the domain of your study aim. However, to genuinely contribute with an article that is expected to synthesize previous contributions in the field and point our further research, i deem that the authors need to do some more work, particularly to sharpen the current manuscript.”

R: Thanks for this suggestion. The authors have tried to capture deeper ideas about the development of the research gap, developing this issue more clearly in the Introduction.

“The last paragraph of the Introduction must be done a summary (resume) of the paper, i.e., a clear idea about what will be studied in the paper. I could not identify this. The last paragraph's purpose in the Introduction section is to summarize the main points, restate the paper's main idea, and show how the paper statements were proven.”

“A brief literature review or a brief theoretical background section is required to define the study's main concept and explain. The current conceptual model does not explain well which theory the author applied. Thus, I suggest that this study   Considering that so much research has already been done within these areas, the paper needs a stronger and more focused approach and a more precise positioning. If so, it is essential to discuss the state of knowledge within these areas in much more detail, previous insights and results that this study may relate to, definitions of the main concept.”

R: Thanks for this suggestion. A brief literature review was conducted and a theoretical background section was created to define and explain the main concept of the study, in order to be able to better explain the conceptual model and its relationship with the theory presented.to within these areas in much more detail, the previous knowledge and the results to which this study may be related, the definitions of the main concept.

“Implications for future research may also be included in the conclusion at the end.”

R: Thanks for the suggestion. The authors demonstrated the implications for future research, namely the importance of nursing research and the role of nursing professionals during a pandemic.

“Extensive editing of English language and style required.”

R: Thanks for the suggestion. The authors have done extensive editing of the English language script throughout the article.

Thank you for your helpful comments and suggestions which have contributed to the improvement of this paper.

Round 3

Reviewer 1 Report

- I appreciate the novelty of the author's contributions, but still, authors need to grasp deep ideas on developing the research gap in the Introduction. The manuscript concerning the presentation of ideas shows improvement, but there are still some sentence structure mistakes and develop more clear research gap with the support of previous published studies in the domain of your study aim. However, to genuinely contribute with an article that is expected to synthesize previous contributions in the field and point our further research, i deem that the authors need to do some more work, particularly to sharpen the current manuscript.

-The last paragraph of the Introduction must be done a summary (resume) of the paper, i.e., a clear idea about what will be studied in the paper. I could not identify this. The last paragraph's purpose in the Introduction section is to summarize the main points, restate the paper's main idea, and show how the paper statements were proven. I think the paper will considerably improve and be a highly cited article. See for example

Organizational Collaborative Culture as a Source of Managing Innovation, World Applied Sciences Journal, 24 (5):582–587. ISSN: 1818–4952

Work satisfaction Aspects in Academics: An Empirical Study.World Applied Sciences Journal 28 (12): 2193–2201

-A brief literature review or a brief theoretical background section is required to define the study's main concept and explain. The current conceptual model does not explain well which theory the author applied. Thus, I suggest that this study   Considering that so much research has already been done within these areas, the paper needs a stronger and more focused approach and a more precise positioning. If so, it is essential to discuss the state of knowledge within these areas in much more detail, previous insights and results that this study may relate to, definitions of the main concept.

Author Response

We appreciate your time and effort in reviewing our paper and greatly value your constructive comments. We respond to your remarks below. Thank you.

“I appreciate the novelty of the author's contributions, but still, authors need to grasp deep ideas on developing the research gap in the Introduction. The manuscript concerning the presentation of ideas shows improvement, but there are still some sentence structure mistakes and develop a more clear research gap with the support of previously published studies in the domain of your study aim. However, to genuinely contribute with an article that is expected to synthesize previous contributions in the field and point our further research, I deem that the authors need to do some more work, particularly to sharpen the current manuscript.”

R: Thanks for this suggestion. However, the authors think they have captured more profound ideas about the development of the research gap and have placed these same developments in the Introduction as requested last time.

On the other hand, the authors cannot find other errors in the sentence structure and development of that same research gap. Moreover, the authors believe in having carried out the proper synthesis of the previous contributions, also pointing out the originality of our work.

However, if you cannot verify these same changes, please point out clearly where we should make the proper changes.

“The last paragraph of the Introduction must be done a summary (resume) of the paper, i.e., a clear idea about what will be studied in the paper. I could not identify this. The last paragraph's purpose in the Introduction section is to summarize the main points, restate the paper's main idea, and show how the paper statements were proven.”

“A brief literature review or a brief theoretical background section is required to define the study's main concept and explain. The current conceptual model does not explain well which theory the author applied. Thus, I suggest that this study   Considering that so much research has already been done within these areas, the paper needs a stronger and more focused approach and a more precise positioning. If so, it is essential to discuss the state of knowledge within these areas in much more detail, previous insights and results that this study may relate to, definitions of the main concept.”

R: Thank you for your suggestion. In the last submission, the authors conducted a more in-depth literature review about the theory presented in the conceptual model. The authors even added a new theoretical point to the original article. In addition, the authors further explained the relationships between the theory under discussion and the dimensions understudy, having discussed the already known approaches to this theory and these dimensions, also mentioning what had not been addressed in any other study. This new approach also allowed us to perceive the original nature of our research. Thus, we do not believe that there is much more to be addressed in this article, insofar as the authors do not intend this study to be an extensive literature review of the theory in question, as there are already other previous studies that have used this same approach. The authors think they have managed to fill this gap that the original article presented, and you very correctly pointed that out.

Thank you for your helpful comments and suggestions which have contributed to the improvement of this paper.
